# Identification of Plant Compounds with Mass Spectrometry Imaging (MSI)

**DOI:** 10.3390/metabo14080419

**Published:** 2024-07-30

**Authors:** Nancy Shyrley García-Rojas, Carlos Daniel Sierra-Álvarez, Hilda E. Ramos-Aboites, Abigail Moreno-Pedraza, Robert Winkler

**Affiliations:** 1Unidad de Genómica Avanzada, Cinvestav, Km. 9.6 Libramiento Norte Carr. Irapuato-León, Irapuato 36824, Mexico; shyrley.garcia@cinvestav.mx (N.S.G.-R.); carlos.sierra@cinvestav.mx (C.D.S.-Á.); hilda.ramos@cinvestav.mx (H.E.R.-A.); 2Leibniz Institute of Vegetable and Ornamental Crops (IGZ) e.V., Theodor-Echtermeyer-Weg 1, 14979 Großbeeren, Germany; 3Institute of Biodiversity, Friedrich Schiller University Jena, Dornburger-Str. 159, 07743 Jena, Germany

**Keywords:** plant metabolomics, mass spectrometry imaging, compound identification

## Abstract

The presence and localization of plant metabolites are indicative of physiological processes, e.g., under biotic and abiotic stress conditions. Further, the chemical composition of plant parts is related to their quality as food or for medicinal applications. Mass spectrometry imaging (MSI) has become a popular analytical technique for exploring and visualizing the spatial distribution of plant molecules within a tissue. This review provides a summary of mass spectrometry methods used for mapping and identifying metabolites in plant tissues. We present the benefits and the disadvantages of both vacuum and ambient ionization methods, considering direct and indirect approaches. Finally, we discuss the current limitations in annotating and identifying molecules and perspectives for future investigations.

## 1. Mass Spectrometry Imaging of Plants

Plant metabolites have been extensively studied as a source of bioactive compounds for different industries. Plant biology has sought to elucidate when, where, and which secondary metabolites act as chemical mediators between plants and their surrounding environment [1].

Gas chromatography (GC) and liquid chromatography (LC) coupled to mass spectrometry (MS) are routinely used analytical methods for studying plant metabolites. However, the precise location within the tissues remains unknown due to the limitation of sample extraction [2].

Mass spectrometry imaging (MSI) is a technique that generates a snapshot of the distribution of molecules in biological tissue at a specific time. MSI permits the visualization of a diverse range of compounds in a single experiment. MS is a universal method, and in contrast with other techniques like antibody-based strategies, MS allows for the exploration of a more comprehensive chemical profile within the same experiment. Consequently, MSI represents an exciting opportunity to refine our knowledge of plant physiology.

Previous reviews on MSI in plants have concentrated on several key aspects, including sample preparation for MSI [3], the main ionization methods [4], compound classes [5], and perspectives on MSI in plant science [6,7].

Nevertheless, compound identification from MSI datasets can be challenging because each sampled spot may contain multiple overlapping signals, which are difficult to separate by experimental or data processing methods. As a result, MSI is more a quantification than an identification technique.

Here, we review the state-of-the-art plant MSI, levels of confidence for metabolite identification, and experimental and computational strategies that can be used in different stages of MSI projects to enrich, separate, and identify metabolites.

An MSI setup, independently from the manufacturer, consists of a sampling plate on which the tissue is positioned, an ionization/desorption source, a mass analyzer, and a detector.

## 2. Overview of Ionization Sources Used for Imaging Plant
Tissues

Over the past two decades, new ionization sources have expanded the range of detectable compounds. The joint goals are to allow for direct analyses and enhance sensitivity [3]. Among the numerous ionization sources, we can distinguish two principal categories depending on the pressure conditions of the ionization/desorption source, i.e., analysis under vacuum or ambient conditions.

Matrix-assisted laser desorption ionization (MALDI) and secondary-ion mass spectrometry (SIMS) represent the most popular ionization techniques under vacuum conditions [8]. SIMS uses highly energetic primary ions for ionization, resulting in a high degree of fragmentation and making this technique less attractive for identifying unknown species [9]. In MALDI, analytes are co-crystalized with a chemical matrix. The matrix is a small molecule that absorbs the energy from a highly energetic ultraviolet (UV) or infrared (IR) laser, resulting in lower fragmentation. A significant challenge in MALDI imaging experiments is the high number of interference signals from the chemical matrix, which often overlap with the metabolites of interest [10]. Another critical aspect is sample preparation, which consists of a multi-step process that may result in the delocalization of the analytes and, subsequently, erroneous interpretation of the results. MALDI was used for the first time in plant science in 2005 to detect and image agrochemicals in soybean plants [11].

In 2007, the colloidal graphite-assisted laser desorption ionization (GALDI) technique was introduced as an alternative to the standard MALDI matrixes. Due to its hydrophobic nature, graphite is more compatible with plant material. GALDI was used to image small molecules in strawberry and apple slices. Fatty acids and flavonoids were analyzed in negative mode in the femtomole range. The molecules were detected directly and subsequently identified by comparing the MS and MS/MS spectra with those of standards [12,13].

In 2009, a matrix-free approach was presented to analyze flavonoids in the model plant *Arabidopsis thaliana*. This technique was introduced as laser desorption ionization (LDI) and is ideal for UV-absorbing compounds [14]. In the same year, Harada et al. presented a mass-microscopic atmospheric-pressure (AP) LDI [15]. A UV laser under ambient conditions was combined with an optical microscope to image fresh ginger rhizome sections. The imaging system was coupled to a quadrupole ion trap time-of-flight (QIT-TOF) instrument, which facilitated identification since tandem MS could be compared to the authentic standard. Another approach used an LDI to image trichomes from wild tomatoes (*Solanum habrochaites*). In this report, the authors used a carbon-substrate-based method to transfer the trichomes from the tissue to the carbon slide, resulting in images with high spatial resolution [16]. LC-MS confirmed the identification of metabolites. Postsource decay LDI mass spectra were acquired to verify the metabolite structures further.

In MALDI-2, an additional laser improves the sensitivity and detection of low-abundant compounds by offering a second ionization stage [17]. MALDI-2 was used for imaging apple (*Malus domestica*) sections. Compared with conventional MALDI, sugars and phenolic compounds were detected with an increment of two orders of magnitude.

AP-MALDI was introduced in 2000 to overcome the sublimation of matrixes under high-vacuum conditions [18]. In 2007, AP-MALDI was applied for the first time to study plant tissues. Initially, a mid-IR laser was used to visualize sugars and citric acid on strawberry skin [19]. Using an IR laser eliminates the necessity of an external chemical matrix because the water content in the tissue serves as a matrix for absorbing the laser energy.

The AP scanning microprobe (S)MALDI was developed for high lateral resolution [20]. A single laser pulse per pixel was applied for imaging peptides. In 2014, this system was used to image plant tissues; the rhizome of *Glycyrrhiza glabra* was investigated, and two isobaric saponins were mapped with high lateral resolution. Further MS/MS analysis confirmed their identity [21].

Multiple ambient ionization mass spectrometry (AIMS) techniques have also been developed. Their popularity relies on their ability to perform direct analysis, reducing or eliminating sample preparation. AIMS analyses are conducted under natural conditions where plant tissues do not undergo significant changes. The AIMS techniques can be classified based on the desorption/ionization mechanism. This classification includes spray-based, plasma-based, and coupled techniques [22].

Desorption electrospray (DESI) is the most common spray-based technique for MSI under ambient conditions. DESI was presented in 2004 [23] and later coupled to imaging platforms [24]. DESI does not require a matrix application; instead, it relies on an electrically charged solvent that impacts the surface of the tissues. The analyte extraction depends on the solvent used, the sample’s complexity, and the setup’s geometry [25]. DESI has been used for direct and indirect analyses of plant tissues. In direct mode, biological tissues are mounted on the sampling stage without prior preparation. The first example of direct DESI imaging analysis was presented in 2009, where the macroalga *Callophycus serratus* was studied; bromophycolides were mapped before and after mechanical damage. The identification of the secondary metabolites was corroborated by comparison of the extracts using LC-MS analysis with pure standards [26]. However, direct DESI imaging in plant tissues has a main obstacle: the penetration of the cuticular wax layer. To overcome this problem, Janfelt’s group presented an indirect DESI imaging technique in 2011. The imprinting of the sample by using a micro-porous Teflon surface extracts the compounds from their natural matrix while maintaining the spatial integrity of the sample. Leaves and petals of *Hypericum perforatum* and *Datura stramonium* leaves were investigated, and compound identification was achieved with tandem MS on the imprints [27]. A couple of months later, Cooks’ group presented the same approach to investigate the catabolic products from chlorophyll in *Cercidiphyllum japonicum* leaves. Tandem MS using collision-induced dissociation (CID) verified the identity of the catabolites [28].

Direct real-time analysis (DART) was the first ambient technique based on plasma [29]. DART generates plasma by using direct current discharge and helium flow. The temperature of plasma impacting the sample surface ranges between 250 °C and 350 °C. The lack of a defined plasma jet and its high temperature makes DART impractical for imaging applications. Low-temperature plasma (LTP) MS was introduced in 2008 [30]. Alternating high voltage, high frequency, and low gas flow produce dielectric barrier discharge. The temperature at the sample surface is maintained around 30 °C. A double dielectric barrier probe was presented in 2013 [31]. This LTP with a defined plasma beam diameter and controlled temperature is suitable for directly analyzing plant tissues. Soon after, LTP was used for imaging the distribution of capsaicin and other small metabolites in a cross-section of chili (*Capsicum annum* Jalapeño pepper) fruit [32]. In this example, the spatial resolution was limited to 1 mm; however, it was sufficient to demonstrate that capsaicin is distributed and limited to specific fruit compartments. A 3D-printed version of the LTP probe with a plasma jet diameter of about 200 μm can be used for LTP MSI with improved lateral resolution and *in vivo* analysis of plants, such as nicotine biosynthesis in tobacco [33,34].

Coupled techniques separate the desorption and the ionization of the analytes. Surface sampling with a laser has several advantages for imaging applications. Lasers can be optically focused to provide high spatial resolution [35]. Laser ablation electrospray ionization (LAESI) was presented in 2007 as a coupled technique for directly analyzing biological samples [36]. Tissue-specific metabolites from *Tagetes patula* seedlings were reported and tentatively identified based on the combination of accurate masses, database, and isotopic distribution and by using tandem MS. The year after, an imaging study showed the capabilities of LAESI for plant analysis [37]. Leaves from *Aphelandra squarrosa* were imaged, and characteristic metabolites from green and yellow areas were identified by using tandem mass spectrometry. MALDESI was developed as a combined technique, including applying a matrix followed by a laser to ablate the area and a second ionization process using DESI [38]. The desorption ionization mechanism is similar to LAESI; for IR-MALDESI, applying an ice layer as a matrix showed an increase in the ion intensity by an order of magnitude [39].

In 2020, IR-MALDESI was used to image cherry tomatoes’ metabolites [40]. The authors employed discovery-driven and literature-driven methods for identifying compounds in tomato MSI. The discovery-driven method involves analyzing the spatial distribution of selected ions and correlating other ions with the same distribution pattern. After assessing the distribution of the most relevant ions, a literature-driven method was used, and the ions were searched against the literature and database resources. For example, thirty-five structural isomers were found for the ion 273.0757 *m*/*z*, and among these, naringenin chalcone was selected as the most likely metabolite. Laser ablation atmospheric-pressure photoionization (LA-APPI) is another approach presented in 2014 to investigate the distribution of active compounds in sage (*Salvia officinalis*) leaves [41]. The tentative assignment of the primary ions was based on previous reports.

Plasma-based techniques have also benefited from adding an extra desorption source and have been used for imaging plant tissues. In 2008, Nd:YAG laser ablation (LA) was coupled to a flowing atmospheric-pressure afterglow (FAPA), demonstrating the capabilities for imaging. In this report, the authors doped celery stock with caffeine and imaged it [42], concluding that adding a laser improves the detection of compounds and makes imaging analysis possible. In 2014, plasma-assisted laser desorption ionization mass spectrometry (PALDI-MS) was presented by using a DART ionization source and a 532 nm laser [43]. The study demonstrates the non-uniform distribution of the active components baicalein and wogonin in *Radix Scutellariae* root. The accurate mass-to-charge ratio confirmed the identity of these compounds. In 2017, a similar study was presented using a 213 nm Nd:YAG solid-state UV laser and DART; this technique was called laser ablation direct analysis in real time (LADI). LADI-MS was used to image the spatial distribution of the alkaloid biosynthesis products in the *Datura leichhardtii* seed [44]. MALDI-MS/MS was used to corroborate the identity of the observed masses. In 2019, another plasma-based approach was presented; a UV diode laser was coupled to LTP to improve the desorption of less volatile compounds and delimitate the area of analysis [45]. This coupled technique, LD-LTP, was used to image mescaline in the cactus San Pedro (*Echinopsis pachanoi*), nicotine in tobacco (*Nicotiana tabacum*) seedlings, and tropane alkaloids in jimsonweed (*Datura stramonium*) fruits and seeds. This technique demonstrated its flexibility for the analysis of macroscopic and mesoscopic samples. The identification of targeted molecules was conducted by using MS/MS. In 2024, a similar concept was put together by using a 532 nm laser ablation system followed by dielectric barrier discharge ionization (LA-DBDI) to study pesticide uptake and translocation in tomato plants [46].

The imaging techniques described here are examples of MSI under vacuum and ambient conditions, as well as coupled techniques. Now, we will review methods for elucidating and validating plant molecules from MSI experiments.

## 3. Plant Compound Elucidation in Mass Spectrometry Imaging
(MSI)

The metabolomics standards initiative defined four plus one metabolite identification levels, reflecting their experimental support [47,48]. These levels were revised, and now, using five levels of confidence is recommended for both untargeted metabolomics and MSI experiments [49,50,51].

We represent them in reverse order with examples of applications from the lowest to the highest confidence level:**Level 5—Exact mass of interest:** The raw data contain defined *m/z* signals that can be mapped to the sampled surface. With sufficient analytical resolution, it can be assumed that the *m/z* features correspond to unique compounds. Of course, isobaric molecules cannot be distinguished. Features are not identified; however, quantitation and statistical analyses for finding regions of interest (ROIs) or potential biomarkers are possible.**Level 4—Molecular formula:** High-resolution mass spectrometry (HR-MS) data, fragmentation experiments, and isotopic patterns permit calculating the chemical sum formula. The results can be compared with databases to find a possible match.**Level 3—Tentative structure:** Based on HR-MS data, tandem MS directly from tissues, in-source decay spectra, isotope distribution, and databases. More than one compound can be explained by using the available data. This level requires complementary information, such as multimodal imaging techniques, fluorescence microscopy, IR spectroscopy, immunolocalization, chemical staining for functional groups, tissue extracts, and subsequent analysis using GC-MS and LC-MS.**Level 2—Probable structure:** Further refinement leads to a single structure candidate. The results obtained in level 3 are assessed by using expert knowledge, biological context, and bioinformatic analyses. For example, genome analyses and chemoinformatics can reveal theoretically possible metabolites.**Level 1—Confirmed structure:** Unequivocal three-dimensional chemical structure identification. Requiring at least two independent and orthogonal methods should provide different types of information and not be affected by the exact source of error. For example, Nuclear Magnetic Resonance (NMR) supports structural studies, and isotopic labeling techniques enable tracing the path of a molecule through a reaction or a metabolic pathway. An authentic standard is required; in MSI, it is common practice to spike it into a replicated biological tissue.

Figure 1 and Table 1 illustrate the confidence levels for mass spectrometry imaging experiments and suggest methods for compound identification that can be combined with mass spectrometry imaging (MSI).

Reaching level 1 for proper chemical identification is challenging for several reasons, such as the need for reference standards, low signal intensities and low abundance of the compound of interest, and the overlap of *m/z* features. Therefore, it is crucial to define the biological question and, based on this, identify a suitable experimental strategy. The following section discusses the steps commonly used to perform MSI experiments.

## 4. Experimental Steps in MSI

The MSI studies can be divided into the four main steps (Figure 2) listed below:Sample preparation.MSI analysis (data collection).MSI data analysis.Supportive techniques.

Table 2 presents reported MSI studies and used supportive techniques.

### 4.1. Sample Preparation

MSI analysis typically requires extensive preparation to accurately capture the metabolic status of the biological tissue at a specific point and under defined conditions (e.g., seedlings under drought stress vs. control seedlings). Over the past 20 years, the predominant technique for MSI in plant science has been MALDI, followed by the ambient ionization technique, DESI. Ambient ionization techniques require much less sample preparation, and ideally, the biological materials can be analyzed ‘as is’. Sample preparation should consider time preservation, sectioning, and optional chemical treatments, such as matrix application or in situ chemical derivatization.

#### 4.1.1. Sample Preservation

It is essential to have appropriate handling and preservation techniques for the samples to ensure the molecules’ original distribution, abundance, and identity. Variables such as water content, storage time, and temperature to quench metabolism have to be considered. Plant tissues are commonly freeze-dried (e.g., for SIMS analysis) or frozen, sectioned, and stored at −80 °C (e.g., for MALDI or DESI analyses) to prevent enzymatic degradation or analyte diffusion [52]. Freezing biological material with a high water content without further precautions is not recommended. Immediate freezing and storing plant tissues can alter the sample’s initial shape over time due to water sublimation, which causes tissue shrinking and the delocalization of compounds. A strategy to avoid delocalization is to slide tissue sections, mount them in glass slides, vacuum-dry them, and then store samples in falcon tubes with holes vacuum-sealed in plastic bags [3]. Using a method that quenches the metabolism is highly recommended for any biological sample. For example, after being embedded in gelatin and immediately frozen for cryosectioning, leaves may be warmed and vacuum-dried [53].

#### 4.1.2. Sectioning

Once the biological tissue has been adequately preserved, the next step is typically sectioning via a microtome or a cryo-microtome, and the resulting sections are mounted on a flat surface. However, this step depends on the compounds of interest, the selected MSI technique, and the tissue type. For example, for MALDI imaging, conductive glass slides are recommended [8]. The thickness of the tissue sample is also essential. Finer slices improve conductivity in MALDI experiments. Cutting thin tissue slices with high water content is technically challenging. Typically, plant tissue slices are in the range of 50 μm [3]. Thin tissue sections are not always necessary, e.g., for tiny seeds (1–3 mm in diameter). There is a limitation due to the thickness of some seeds, leaves, flowers, and even small roots; therefore, it cannot be sectioned. Thus, the tissues may be studied intact or imprinted [54,55]. In 2015, transversal cuts of maize leaves with a thickness of 10 μm were obtained [53]. Thin tissue slices were also achieved by placing a root between two polystyrene sections and cutting it tangentially [56].

Histology methods for sectioning have been widely adopted for MSI. However, plant tissue fixation, washing, and staining are incompatible with MS-based techniques. Polyethylene glycol should be avoided because of metabolite diffusion and signal suppression [3]. New ambient imaging techniques presented reduce sample preservation and sectioning; for instance, with the use of LD-LTP and LADI, sample preservation is not necessary for tissues with low water content, and the sectioning of the samples can be accomplished with a scalpel [44,45].

#### 4.1.3. Liberation of Plant Cell
Compounds

Adequate sample handling avoids diffusion and degradation through enzymatic processes and light, heat, and atmospheric exposure [3]. The preparation and treatments employed before analysis in plant samples differ from those used in mammalian tissues because of the plant cuticle and the cell wall barriers. Three strategies have been employed to address these challenges.

The first strategy is to use an efficient ionization/desorption source according to the plant tissue. Alternatively, coupled ionization/desorption techniques may be employed; thus, a direct analysis is performed in one or two steps. In this case, a desorption/ablation source penetrates the tissue, facilitating the release of the molecules. Subsequently or simultaneously, an ionization source, for example, a laser, as in LAESI or LD-LTP, is employed to assist the process.

The second strategy involves imprinting a plant tissue section on flat surfaces, such as PTFE (polytetrafluoroethylene) or nylon sheets [27,55]. This approach is considered an indirect technique for MSI. This process avoids the potential interferences caused by the cuticle, cellular wall, and water content. Ifa’s group presented another indirect technique used in plant tissues: blotting assisted by heating and solvent extraction, using a thin-layer chromatography plate to transfer the plant material components [57].

In 2015, Klein et al. presented a similar technique to liberate and expose internal molecules for subsequent MALDI imaging. The method was called ‘fracturing’ and was employed to study plant–bacteria interactions. The authors pressed a vacuum-dried rice leaf and built a sandwich by using tape. The leaf was then passed through a rolling mill. The fractured leaf section was placed on a MALDI target plate with double-sided tape so that it would adhere to it and subjected to matrix application and further MSI analysis [58].

Finally, the third strategy is to remove the plant cuticle, which can be achieved through two methods, e.g., a chemical wash-off, which facilitates the removal of the wax layer from tissues, thereby boosting detectable compounds that would otherwise be undetectable with a direct technique. The analysis of chloroform-dipped *Arabidopsis* leaves was presented in 2008 by Cha et al. [13]. However, this process may also result in the delocalization or washing away of many other compounds, as Tong et al. demonstrated in 2022 [54].

#### 4.1.4. Matrix Application

In MALDI and MALDESI, a chemical matrix is required. Commonly, this matrix is a small organic compound that facilitates the desorption and ionization of the compounds of interest. The matrix choice is analyte-dependent and is crucial for MSI to avoid metabolite diffusion, which is an issue in matrix-based techniques.

Correct matrix application and crystallization are crucial for reliable imaging analysis. The most common methods are the use of an airbrush, sublimation, and the use of an automatic sprayer [59]. In 2014, a comparative analysis of these three popular methods analyzed the distribution of metabolites in *Medicago truncatula* root nodules. The automatic sprayer yielded the highest number of detected metabolites [60]. The second best method was sublimation under reduced pressure and elevated temperatures, which created an even matrix layer. In contrast, using an airbrush depends strongly on the operator and reduces the reproducibility of results. Thus, different options for matrix applications are available. Errors in the matrix choice and application might result in artifacts during the measurement. In addition, the crystal size and homogeneity will directly affect the lateral resolution in the MSI analysis.

##### New Matrixes

The discovery and application of new matrixes are still under active development. Using the so-called proton sponge matrixes is a considerable advance in MALDI, especially for negative ionization mode. Proton sponge matrix is a strong base that can deprotonate acidic analytes in the liquid phase, forming an ion pair [A−H]^−^/[B+H]^+^ [61,62]. The novelty lies in their effectiveness for negative ion mode analysis of several acidic low-molecular-weight compounds. On the contrary, a significant drawback of this strategy is the high sublimation under high-vacuum conditions. Other superbases have been described as being used to overcome the sublimation problem [63]. Shroff and Svatos (2009) used 1,8-bis(dimethyl-amino)naphthalene (DMAN) to analyze plant hormones and crude regurgitate of *Manduca sexta* [64]. In 2013, the first report on plant imaging was presented, investigating roots and root nodules of *M. truncatula* during nitrogen fixation. Results were complemented by using 2,5–dihydroxybenzoic acid (DHB) as the matrix for positive mode. The results showed small- and medium-sized molecules such as organic acids, amino acids, sugars, lipids, and flavonoids [65].

In 2014, Korte and Jin Lee applied the matrix 1,5-diaminonaphthalene (DAN) to a maize leaf section to image phospholipids and low-molecular-weight compounds. This resulted in better efficiency than traditional matrixes and a very low background for negative ionization mode [66].

##### Inorganic Matrixes

GALDI, presented in 2006, offers an alternative to the traditional organic matrixes by using graphite to enhance the detection of organic acids, flavonoids, and oligosaccharides in apple and strawberry tissues [67]. The combination of an organic and inorganic mixture matrix demonstrated, in 2016, a reduction in triacylglycerol (TG) ion suppression. The binary matrix composed of DHB and Fe_3_O_4_ nanoparticles was tested in a germinated maize seed section, showing a wide variety of lipids as well as large polysaccharides, showcasing the synergic effect [68]. In the same year, Ozawa et al. [69] demonstrated using a platinum film as the matrix in the surface-assisted laser desorption/ionization technique. Pt-SALDI was used to analyze and map the distribution of a pesticide containing acephate as a vermicide in a cross-section of soybean cotyledon [69].

As previously described, using inorganic nanoparticles is an alternative to traditional matrixes. The advantage of these matrixes is that these materials do not ionize themselves, thus reducing matrix interferences. In 2020, Shiono et al. demonstrated how the use of nanoparticles in LDI-MSI improved the detection of phytohormones of the nine tested phytohormones from 5/9 to 9/9, compared with traditional MALDI [70].

##### Reactive Matrixes

The use of reactive matrixes in MALDI imaging experiments refers to those chemical substances that can act as matrixes and derivatization agents [62]. Zhang and Gross [71] demonstrated, in 2002, *in situ* derivatization, where oligodeoxynucleotides reacted with the matrix anthranilic acid to form a Schiff base. The adduct upon MALDI then fragmented at the abasic site, revealing its location. *In situ* derivatization is also known as *on-tissue* chemical derivatization (OTCD). Promising OTCD on plant tissue was reported in 2023 by Zemaitis et al. to enhance MALDI-MSI detection [72]. The authors applied 4-(2-((4-bromophenethyl) dimethylammonio) ethoxy) benzenaminium bromide (4-APEBA) to derivatize molecules from soybean nodules and poplar roots. The application of 4-APEBA increased the detection of various compounds, including amino acids and hormones, reducing sugars, aldehydes, carboxylic acids, and others [73]. This chemical reagent represents a promising tool for MSI, as it enables the imaging of the distribution of metabolites of opposite polarities and hydrophobicities [72].

Other reviews [59,62,74,75] have extensively discussed reactive MALDI matrixes.

In AIMS techniques, specifically DESI, the use of reactive solvent sprays has enhanced the analysis of plant tissues [76]. As an example, Li et al. tested a ternary solvent system of chloroform: acetonitrile: water (1:1:0.04), resulting in the detection of very-long-chain fatty acids and other metabolites in leaves and petals of *Hypericum perforatum* [77].

### 4.2. MSI Analysis (Data
Collection)

Mass spectrometry imaging (MSI) reduces the confidence in assigning molecular identities because of the absence of separation methods. Therefore, it is recommended to use high-resolution (HR) or ultra-high-resolution (UHR) MS analyzers. Furthermore, other methods, such as ion mobility spectrometry (IMS), mass fragmentation experiments, and utilizing isotopically labeled standard reagents, can increase the confidence level of compound identification.

This section will discuss these approaches and their respective limitations. Using HR-MS or UHR-MS for imaging can reduce isobaric interferences, improving the identification confidence, but UHR-MS is not a routine approach in the MSI of plants [51].

Recently, the combination of MALDI with Fourier transform ion cyclotron resonance (FT-ICR) for MSI has been reported. As FT-ICR is considered a technique of UHR, combining FT-ICR with MALDI can potentially increase the confidence level in plant MSI. However, higher mass and spatial resolutions require longer measurement times and greater computational power [78].

Another critical element is the spatial resolution for MSI, which is crucial to answering a biological question. For example, a high spatial resolution allows for the exploration of the metabolites at the single-cell and organelle level. In 2009, Holscher et al. reported the distribution of secondary metabolites in *Arabidopsis thaliana* and Hypericum species at the single-cell level, using LDI, achieving a lateral resolution of 10 × 10 μm [14]. This resolution is sufficient for sampling at the plant cell level (10–100 μm) [79]. Harada et al. (2009) demonstrated the MSI of volatiles at the organelle level of the ginger rhizome by LDI-QIT-TOF [15].

In addition, the precision of the sampling stage and synchronization with the mass analyzer are crucial for the correct assembly of the MSI datasets.

An exciting advancement in MSI involves integrating ion mobility spectrometry (IMS) into the mass analyzers. IMS offers an additional analytical dimension by separating isobaric compounds based on their 3D structure, thereby facilitating the separation of interesting analytes from similar structural isomers.

Li et al. [80] integrated IMS with LAESI-MSI to image metabolites in leaves of *Pelargonium peltatum*. Through this approach, the authors discovered structural isomers colocalized. Likewise, Zhang et al. [81] combined IMS with DESI-MSI to detect endogenous and exogenous isomers of auxin derivates from *A. thaliana* seedlings.

Adding a second separation method to MSI, such as IMS, can significantly improve the identification of plant metabolites by facilitating *in situ* isobaric compound separation. While the integration of IMS-MSI in plant studies remains relatively unexplored, it is a promising avenue of research. Such a combination introduces a fourth dimension to MSI, i.e., *m/z*, ion intensity, spatial coordinates, and drift time.

A drawback of IMS is its limitation to targeted analysis and low resolution. Thus, more research is needed to explore its advantages for plant MSI.

### 4.3. MSI Data Analysis

Mass spectrometry data analysis workflows require the following steps [82]:Raw data import/export and conversion (if necessary).Spectrum preprocessing.Feature analysis.Statistics and data mining.Integration and interpretation.

MSI data analysis is similar but focuses on visualizing metabolite localization and abundance. Usually, an MSI analysis contains several hundreds or even thousands of features, and depending on the research question, identifying these ions can be less or more complex. The steps for analyzing MSI data are more directed toward visualizing metabolite distribution and biological interpretation. The first step involves creating the spatial distribution of any given mass-to-charge value or range.

Commercial instruments with MSI setup have vendor-specific software for data processing. In addition, academic instrument developers and programmers are creating in-house software to analyze the data recorded with their own or any given MSI instrument [83]. The vendor’s software requires purchasing a license, which can be costly and sometimes lacks the functions needed to answer a bioanalytical question. Therefore, MSI data processing workflows with open-source software are attractive for academic researchers. File conversion from instrument data to community file formats is usually necessary. The standardized data format for mass spectrometry imaging data imzML has facilitated data management and MSI data exchange [84]. Many vendors’ software applications can now read imzML files (MassLynx, FlexImaging, and Imaging For Windows) or provide an option to export imzML files (http://ms-imaging.org/imzml/software-tools/, accessed on 25 July 2024). In addition, several open-source MSI programs provide data conversion [85] and support for imzML files [86,87,88,89,90,91].

The open-source program RmsiGUI integrates the control of a robotic imaging platform and the compilation of imzML [83]. Software like MSiReader v2.0 or RmsiGUI (R package) allows for the overlay of optical and ion images [83,87].

The analysis of MSI data often includes visualization of metabolite distribution, quantitation, or other statistical features, including principal component analysis (PCA). Free and vendor software, such as MSiReader and msiQuant v.2.0.1.15 [87,92], are available.

Accurate mass filtering (e.g., HRMS) and a tandem MSI analysis support the identification of compounds from MSI experiments. An R package, rMSIFragment, recently included a function to improve lipidomics annotation considering in-source fragmentation [93]. In 2023, Wadie et al. developed METASPACE-ML, a machine learning model for METASPACE which includes false rate discovery (FDR) calculation, to improve reliability in metabolite annotation [94]. Most data used to train the model were MALDI-based and are still in development. However, tools like METASPACE-ML and rMSIfragment will increase the confidence for future MSI data annotation and identification.

Several free MSI data processing programs are implemented in the free statistical computing and graphics programming language R. R has an active user community and many additional data evaluation and visualization packages. However, R is an interpreter and, therefore, relatively slow. Reading imzML with the Julia language resulted in 100 times faster loading speeds than R, demonstrating the potential for the future data mining of massive MSI datasets [95].

In general, MSI software has options for visualization, normalization, or multivariate analysis, but only a few applications include a molecular annotation/identification function, such as LipostarMSI [96], METASPACE [94], Scils^TM^ Lab, and Cardinal [97]. Recently, the MZmine 3 project presented an alternative function for MSI analyses. The possibility of importing LC-IMS-MS and IMS-MSI datasets allows for the analysis of both datasets simultaneously. The resulting data matrix combines both results, improving compound annotation when using in-house libraries, open MS libraries, and using the function Feature-Based Molecular Networking (FBMN) to upload it to the Global Natural Products Social Molecular Networking (GNPS) (https://gnps.ucsd.edu/, accessed on 25 July 2024) Wang et al. [98].

PRIDE (https://www.ebi.ac.uk/pride/, accessed on 25 July 2024) [99] and METASPACE (https://metaspace2020.eu/, accessed on 25 July 2024) provide public repositories for uploading MSI files to facilitate collaboration in the research community.

Using dedicated databases for plants (e.g., PlantCyc (https://www.plantcyc.org, accessed on 25 July 2024)) increases the confidence in compound identification compared with using generic databases [100].

Integrating MSI with multi-omics is a promising and valuable approach to understanding the gene–metabolite relationship and discovering novel gene-associated metabolites [101]. In 2020, Dong et al. demonstrated that coupling MSI with RNA interference, gene silencing, agro-infiltration, or samples derived from plant natural variation could spatially map an entire metabolic pathway [101].

Despite the advances in software for MSI data processing, there is still much left to do. Currently, MSI data analysis is mainly manual, which is time-consuming and error-prone. Thus, the development of automated analysis of MSI datasets is required. For a more detailed revision of MSI software, check Weiskirchen et al., 2019 [89].

### 4.4. Supportive Techniques

Even with HR-MS instruments, identifying and confirming structural isomers is complicated [102]. Therefore, verifying compounds identified in an MSI experiment usually requires supportive techniques. For fragmentation studies on the features of interest, using the same ionization source and references is recommended. Fragmentation patterns can be compared with or without spiking the tissue with a standard compound.

ROIs can also be extracted from the tissue and analyzed with LC-MS [54,102]. Recovering the ROIs is also handy when reference standards are unavailable [15,55].

Additional techniques are often used to analyze the analytes’ chemical and physical properties, e.g., using HPLC coupled to diode array detection (DAD).

**Table 2 metabolites-14-00419-t002:** Overview of strategies for mass spectrometry imaging (MSI) and supportive techniques used in the identification of plant metabolites. Abbreviations are listed below.

Chemical Class	Analyte	MSI Techn.	Orthol. Methods	Complementary Techn.	ID Level	Refs.
Phenolic compounds	Resveratrol, pterostilbene, and stilbene phytoalexins	LDI and MALDI	HPLC-DAD	Fluorescence imaging (macroscopy) and confocal fluorescence microscopy	Level 2	[103]
Volatiles and phenolic compounds	Gingerol and terpenoids	AP-LDI	AP LDI MS/MS	Optical microscopy	Level 2	[15]
Flavonoids	Kaempferol, quercetin, and isorhamnetin	LDI	AP-MALDI and CID (TOF/TOF)	-	Level 2	[14]
Flavanones	Baicalein, baicalin, and wogonin	MALDI	MALDI-Q-TOF-MS	Optical microscopy	Level 2	[104]
Phenolic compounds and carbohydrates	Jasmone, hexose sugars, salvigenin, flavonoids, and fatty acyl glycosides	DESI-MSI	FS FAAS	-	Level 2, level 3, and level 4	[105]
S-glucosides	Glucosinolates	MALDI, LAESI	ESI (chip-ESI)	-	Level 2	[106]
Phenolic compounds and carbohydrates	Salvianolic acid J	DESI	LC-MS	-	Level 3	[54]
Organic acids, phenolics, and oligosaccharides	Ascorbic acid, citric acid, palmitic acid, linoleic acid, linolenic acid, oleic acid, apigenin, kaempferol, ellagic acid, quercetin, apigenin, fructose, glucose, and sucrose	MALDI, GALDI	-	-	Level 2	[67]
Amino acids, phenolic compounds, and lipids	Indoxyl, clemastanin B, isatindigobisindoloside G, gluconapin, guanine, adenine, adenosine, sucrose, histidine, lysine, arginine, proline, citric acid, malic acid, and linolenic acid	MALDI	DESI-Q-TOF	-	Level 2	[107]
Hydrocarbons and flavonoids	C29 alkane, kaempferol–hexose, and quercetin–rhamnose	MALDI	DESI-MS, LAESI-MS, SIMS	-	Level 2	[108]
Glycoalkaloids and anthocyanins	Tomatidine, α-tomatine, and dehydrotomatine	MALDI	LC-MS/GC-MS	Electron microscopy imaging	Level 1	[101]
Fatty acid and amino acids	Palmitic acid, stearic acid, oleic acid, inositol, β-Alanine, and tomatidine	MALDI	-	RT-qPCR	Level 2	[109]
Organic acids	Citrate, malate, succinate, and fumarate	MALDI	UPLC-HRMS/MS	-	Level 1	[55]
Anthocyanins	Choline and pelargonidin	MALDI, SIMS	MALDI-MS/MS	Optical microscopy	Level 2	[110]
Lipids	Cuticular lipids	MALDI	GC-MS	-	Level 5	[111]
Terpenoids and diterpenoids	Vitexilactone, vietrifolin D, and rotundifuran	MALDI	GC-MS	-	Level 3	[112]
Lipid droplet associated protein	Wax ester and triacylglycerol	MALDI	-	Confocal micrographs of LDAP	Level 4	[113]
Nitrogenated and phenolic compounds	Cocaine, cinnamoylocaine, benzoylecgonine, etc.	MALDI, LDI	ESI	-	Level 4	[114]
Organic acids, carbohydrates, flavonoids, and lipids	Nobiletin, phenylalanine, trans-Jasmonic Acid, quinic acid, ABA, etc.	DESI	LC-MS/MS	-	Level 3	[115]
Triacylglycerol and phosphatidylcholines	Palmitic acid, vaccenic, linoleic, and α-linoleic acids	MALDI	NMR, ESI	-	Level 1	[116]
Phytohormones	Abscisic, auxin, cytokinin, jasmonic acid, and salicylic acid	PALDI	MALDI	-	Level 2	[70]

Microscopy can be supported by staining, immunolabelling, and label-free techniques. These labeling methods allow for the detection of specific molecules in plant tissues and provide information about the location of enzymes, lipids, carbohydrates, and other molecules.

Dyes like Congo Red or Calcofluor White (CW) stain carbohydrates such as β-(1→4)-glucans such as cellulose, callose, xyloglucans, and chitin [117,118].

In contrast, label-free imaging techniques use novel imaging technologies that do not require stains or fluorescent markers for visualizing plant cell organelles and structures. Ultraviolet microscopy enables the visualization of lignin and other aromatic compounds within plant tissues.

In 2017, Becker et al. [103] demonstrated the efficacy of combining other imaging techniques with MSI. The researchers employed MALDI and LDI for MSI and confocal fluorescence microscopy (CLSM) to study the distribution of phenolic compounds (stilbene phytoalexins, namely, resveratrol, pterostilbene, piceids, and viniferins) in grapevine leaf (*Vitis vinifera*). Moreover, the authors supported their findings in MSI by using CLSM and HPLC-DAD techniques to detect, corroborate, and quantify the phenolic compounds. CLSM produces high-resolution images without damaging the sample, thus allowing for the acquisition of fluorescent images at the organ, tissue, and multicellular levels. By using LDI for MSI (266 nm laser) in positive mode, they could detect, characterize, and localize metabolites such as stilbenes without matrix application. HPLC with a diode array detector (DAD) added quantitative data and validated stilbenes’ compound identification.

Dong et al., in 2020, not only used MSI but complemented MSI information with different reverse genetics approaches to elucidate gene function [101]. They highlighted the potential of integrating MSI with reverse genetics techniques to elucidate gene functions by analyzing the spatial distribution of metabolites and correlating the MSI results with gene expression. The study employed virus-induced gene silencing (VIGS), an RNA-mediated reverse genetics technology, to downregulate endogenous genes and analyze their functions and was complemented with liquid chromatography–mass spectrometry (LC-MS) and gas chromatography–mass spectrometry (GC-MS). In addition, the study employed scanning electron microscopy (SEM) as a complementary imaging technique.

The continued development of multimodal imaging workflows integrating MSI with super-resolution fluorescence microscopy, electron microscopy, spectroscopic imaging techniques like Raman and FTIR, and other emerging methods holds tremendous potential. Innovative multimodal image acquisition, data fusion algorithms, and open data repositories will be critical to fully exploiting the power of multi-scale, multimodal chemical imaging in both fundamental plant biology research and applied areas like food science, bioenergy, and phytoremediation.

## 5. Conclusions and Outlook

The mass spectrometry imaging (MSI) of plants has become a routine method for plant research. However, the unequivocal identification (ID level 1) of chemical structures in MSI data is still challenging. Depending on the bioanalytical question, we suggest a targeted MSI strategy for specific compounds or the statistical analysis and data mining of *m*/*z* features with lower ID confidence (levels 2–5), with subsequent identification of features of interest by using complementary methods.

High-resolution mass spectrometry (HR-MS) analyzers with ion mobility pre-separation facilitate the discrimination of distinct molecule ions. Nevertheless, in most cases, complementary analytical methods will be necessary for identification. In addition, the comparison with authentic reference standards for level 5 identification is not trivial in MSI, and new strategies are needed to overcome practical limitations.

In the near future, we expect advances in integrating instrumental methods with chemoinformatic tools that use advanced algorithms such as machine learning and artificial intelligence to detect and identify biologically important features in MSI datasets. A prerequisite for these developments is the adoption of community file formats, FAIR data sharing and algorithms, and software with permissive, open licenses.

## Figures and Tables

**Figure 1 metabolites-14-00419-f001:**
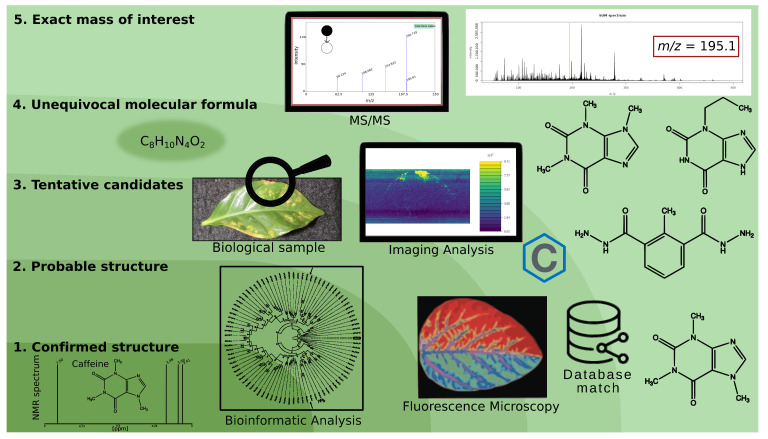
Levels of confidence for the identification of compounds in mass spectrometry imaging.

**Figure 2 metabolites-14-00419-f002:**
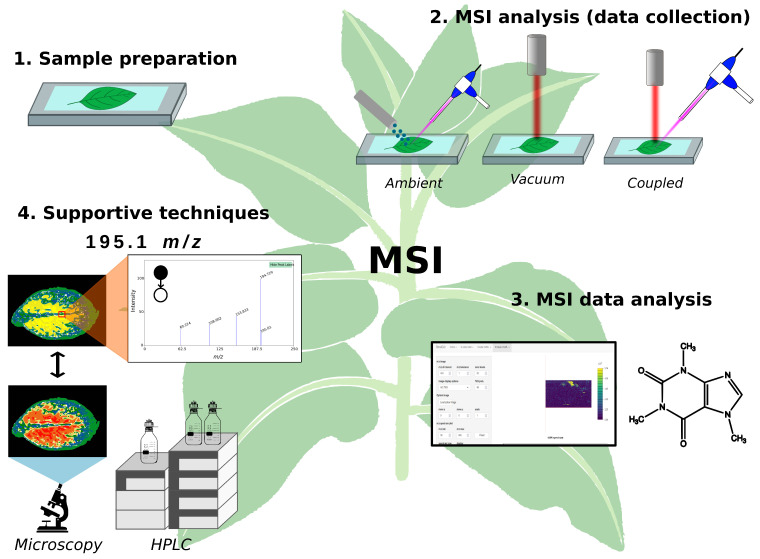
Experimental steps of mass spectrometry imaging (MSI).

**Table 1 metabolites-14-00419-t001:** Identification levels for mass spectrometry imaging (MSI) with examples of applicable methods.

ID Level	Requirement	Mass Spectrometry Imaging
1—Confirmed structure	Unambiguous (3D) structure from at least two independent and orthogonal methods, which refer to methods that provide different types of information and are not affected by the same sources of error and comparison to an authentic reference sample.	Recovery of material from regions of interest (ROIs), which are specific areas selected for detailed analysis; structural studies with orthogonal methods (e.g., NMR and HR-MSn); isotopic label studies, which involve the use of isotopes to trace the path of a molecule through a reaction or a metabolic pathway.
2—Probable structure (single candidate)	Like Level 3, but with only one candidate left.	Filtering results with expert knowledge and bioinformatic analyses (e.g., theoretically possible metabolites from genome analyses and chemoinformatics).
3—Tentative structure (multiple candidates)	HR-MS(n) data match with databases and are congruent with additional experiments and the biological context. Still, more than one compound can be explained with the available data.	High-resolution *m/z* data, direct fragmentation from tissues, in-source decay spectra, and isotope distribution data. Matching with databases and comparison with theoretical spectra. Multimodal imaging (e.g., fluorescence and infrared spectroscopy microscopy; immunolocalization); complementary studies with excisions from regions of interest (ROIs) or complete extractions, using GC-MS and LC-MS; chemical staining for functional groups.
4—Molecular formula	HR-MS(n) and isotopic distribution data of *m/z* features that support the elemental composition of compounds	Calculation of theoretical mass spectra and comparison with experimental data; database matches.
5—Exact mass of interest	*m/z* features are not identified but are unique.	Quantitation and statistical evaluation of *m/z* bins according to their signal intensity.

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
