# Peer review of "Identification of Plant Compounds with Mass Spectrometry Imaging (MSI)"

_metabolites, 2024, doi:10.3390/metabo14080419_

Round 1

Reviewer 1 Report

Comments and Suggestions for Authors

Mass spectrometry imaging (MSI) has emerged as a revolutionary tool in plant science, offering unprecedented insights into the spatial distribution and dynamics of metabolites, lipids, proteins, and other biomolecules within plant tissues. This powerful analytical technique combines the molecular specificity of mass spectrometry with the spatial resolution of imaging, enabling researchers to visualize the complex biochemical landscape of plants in situ. Unfortunately, the application of MSI for plantomics has some drawbacks, such as problems with the identification of analytes and their quantification. However, as technology continues to evolve, MSI is poised to play an increasingly vital role in addressing global challenges related to agriculture, food security, and environmental sustainability. For these points, I consider the manuscript worthy of publication but some major revisions should be made:

1.      The organization of the manuscript does not look well. Please separate introduction and ionization methods sections.

2.      Recently several reviews have been published in the field, please cite at least some of them and add detailed discussion on your review novelty. It is especially important as only less than one third of the cited papers have been published in recent five years.

3.      Though the authors discuss MALDI matrixes there are not any information about reactive matrixes, which are widely used in MSO experiments. In addition, extensive discussion on the matrix application methods is required.

4.      Please add information about the application of neural networks, such as GNPS, for the data interpretation.

5.      The ion mobility techniques should be additionally discussed because of their importance.

6.      Only one example of the derivatization is cited. These approaches sometimes play a crucial role in the detection of the analytes. More examples and extensive discussion are required

Author Response

Reviewer # 1: Mass spectrometry imaging (MSI) has emerged as a revolutionary tool in plant science, offering unprecedented insights into the spatial distribution and dynamics of metabolites, lipids, proteins, and other biomolecules within plant tissues. This powerful analytical technique combines the molecular specificity of mass spectrometry with the spatial resolution of imaging, enabling researchers to visualize the complex biochemical landscape of plants in situ. Unfortunately, the application of MSI for plantomics has some drawbacks, such as problems with the identification of analytes and their quantification. However, as technology continues to evolve, MSI is poised to play an increasingly vital role in addressing global challenges related to agriculture, food security, and environmental sustainability. For these points, I consider the manuscript worthy of publication but some major revisions should be made:

The organization of the manuscript does not look well. Please separate introduction and ionization methods sections.

Recently several reviews have been published in the field, please cite at least some of them and add detailed discussion on your review novelty. It is especially important as only less than one third of the cited papers have been published in recent five years.

Authors: We appreciate your comments. Indeed, several reviews have been published recently in the plant science area. Thanks to your comment, we have included what we consider relevant for our revision. Thanks for your comment on the introduction. We have separated the general introduction from the overview of the different ionization sources used for plant tissue imaging. This organization is much clearer for the readers now.
About the novelty, previous MSI reviews in plants are focused on:
•Sample preparation (https://doi.org/10.3389/fpls.2016.00060).
•Specific MSI ionization methods (https://doi.org/10.1002/bmc.5494).
•Compound classes by MSI (https://doi.org/10.1093/jxb/erad423).
•Generalities and perspectives (https://doi.org/10.1007/s11101-015-9416-2).
This review discusses strategies for identifying and visualizing the spatial distribution of plant metabolites using MSI. We also discuss ionization techniques, supportive techniques, and examples of ID levels.

Reviewer # 1: Though the authors discuss MALDI matrixes there are not any information about reactive matrixes, which are widely used in MSO experiments. In addition, extensive discussion on the matrix application methods is required.

Authors: Thanks for this comment. The review is focused on giving an overview of the different ionization techniques for imaging in plant science and the strategies for annotating the relevant features. One crucial part is sample preparation and the use or not of a matrix. Thanks to your comment, we included a subsection in the revised version, offering the reader a clear view of the traditional, new, inorganic, and reactive matrixes used in plant omics.

Reviewer # 1: Please add information about the application of neural networks, such as GNPS, for the data interpretation.

Authors:Thanks for your comment; the new version includes information about using MZmine 4, Feature-Based Molecular Networking (FBMN), and linking with GNPS for annotation. Even though the annotation rate is still low, we believe that using these new bioinformatic tools together with ML algorithms will make annotation more reliable in the near future.

Reviewer # 1: The ion mobility techniques should be additionally discussed because of their importance.
Authors: We agree; thanks for pointing out ion mobility; the revised version contains an explanation about the use of ion mobility and its usefulness, adding a separation step in MSI analysis. The use of Ion Mobility has increased in the last year. It is handy in the MSI since we add a separation step usually missing in a traditional MSI analysis. However, only some examples in the literature have highlighted the use of such a combination due to the high costs. Still, we are confident that the prices will drop soon and the instruments will become accessible to more laboratories.

Reviewer # 1: Only one example of the derivatization is cited. These approaches sometimes play a crucial role in the detection of the analytes. More examples and extensive discussion are required

Authors:Thanks for your comments. In addition to the matrixes, there is now a more exhaustive explanation of using reactive matrixes for targeted analysis. We hope this information gives readers a better overview of the different options when using MSI.

Reviewer 2 Report

Comments and Suggestions for Authors

Thankyou for submitting your papaer. You have clearly put in a great deal of work. However it's identity is a little muddled. It is not really comprehensive enough for a review - it felt more like a how to guide. It also didn't give a view of the advantages and disadvantages of ambient and vacuum methods of ionisation.

There was some repetition as section 2 (line 203) - Plant compound elucidation in mass spectrometry imaging (MSI) is listed in the text and represented again in figure 1 and again in table 1. Please remove this repetition.

I thnk the overall aim of the paper needs to be addressed - is it a review or is it a how to guide to imaging in plants? This would improve the paper by giving it a better focus.

Author Response

Reviewer #2: Thank you for submitting your paper. You have clearly put in a great deal of work. However it's identity is a little muddled. It is not really comprehensive enough for a review - it felt more like a how to guide. It also didn't give a view of the advantages and disadvantages of ambient and vacuum methods of ionisation.
Authors: Thanks, we slightly rearranged the review. This review intends to give an overview of the different efforts of the scientific community through the years to analyze and map plant material. Vacuum and ambient ionization techniques are included, and we want to highlight the use of the ambient ionization techniques due to their simplicity; at the same time, it is essential to talk about the drawbacks that both methods have. We agree that the review should clearly state the advantages and disadvantages and consider your advice in the revised version. Another point we wanted to make clear in the review is that depending on the biological question, the ionization technique and complementary methods should be selected. We want to give an open and unbiased opinion about both ionization methods.

Reviewer #2: There was some repetition as section 2 (line 203) - Plant compound elucidation in mass spectrometry imaging (MSI) is listed in the text and represented again in figure 1 and again in table 1. Please remove this repetition.
Authors: Thanks for your comment. The table provides a more structured view of the general information provided in the text, and the figure creates a visual demonstration of what we want to transmit. Thus, the table, the text, and the figure are complementary. We appreciate your suggestion, but we would like to keep it as is, considering that some readers are more visual and others want the information summarized in a table format.

Reviewer #2: I thnk the overall aim of the paper needs to be addressed - is it a review or is it a how to guide to imaging in plants? This would improve the paper by giving it a better focus.
Authors: Thanks for your comment. First, the review wants to highlight the importance of imaging in plant science. Then, to showcase the different vacuum and ambient ionization techniques that have been used to study plant material, in this sense, it is straightforward for us to follow a storyline based on what has been done, followed by the how-to, to finally include aspects about the data processing and annotation which is still a bottleneck in general in metabolomics. This review is intended for new users of MSI and, at the same time, may be informative for experts in the area. The latest version contains more details and examples, and we are confident it is now suitable for publication.

Reviewer 3 Report

Comments and Suggestions for Authors

This is a high-quality study which perfectly match the scope of the journal and can be of great interest for the audience since the topic relates to rapidly growing field of the spatially resolved plant metabolomics. The manuscript is well-written and properly organized. I recommend it for publication with a couple of minor corrections:

Line 31. “MSI is more a quantification than an identification technique”. In my opinion, this statement is not quite correct since accurate quantification of analytes in complex matrices requires separation to eliminate matrix effects.  

Line 205. Please, remove the bracket: “plus one) level”

Author Response

Reviewer # 3
This is a high-quality study which perfectly match the scope of the journal and can be of great interest for the audience since the topic relates to rapidly growing field of the spatially resolved plant metabolomics. The manuscript is well-written and properly organized. I recommend it for publication with a couple of minor corrections:
Authors
Thank you for your kind endorsement.

Reviewer # 3
Line 31. “MSI is more a quantification than an identification technique”. In my opinion, this statement is not quite correct since accurate quantification of analytes in complex matrices requires separation to eliminate matrix effects.
Authors
Thanks for your comment. In the MSI of plant materials, more effort has been put into visualization, i.e., the relative quantification of ion signals, compared to annotating unknown compounds. In other words, a targeted approach is more common in MSI experiments. Multiple articles describe methods for improving the quantification of specific chemical classes. However, in most cases, the identification of individual compounds is deficient. Thus, our review states that annotation in MSI is complex due to the lack of separation. The last section of the review gives an overview of supportive techniques to increase confidence in the identification of compounds. The revised version includes ion mobility and more bioinformatic tools that can help with annotation.

Reviewer # 3
Line 205. Please, remove the bracket: “plus one) level”
Authors
Thank you for your comment; we deleted the bracket.

Reviewer 4 Report

Comments and Suggestions for Authors

This review devoted to Mass spectrometry imaging (MSI) for the analysis of plant metabolites is extremely interesting and informative. It describes the main stages of the development of this method and the features at various stages of analysis. The work is well written and reads quickly and easily. It will be especially useful for people who are poorly versed in the field of mass spectrometry to understand the basics of this method. I recommend this work for publication in its current form.

Author Response

Thank you very much for your favorable judgment!

Round 2

Reviewer 1 Report

Comments and Suggestions for Authors

The authors have adressed all issues and the review can be published in my opinion